# Development of X-SIAGA: A disease X and outbreak preparedness intervention for indigenous households in Selangor, Malaysia

Ameerah Su'ad Abdul Shakor[1], Mariam Mohamad[1]*, Khalid Ibrahim[1], Izandis Mohamad Sayed[2]

1 Department of Public Health Medicine, Faculty of Medicine, Universiti Teknologi MARA Sungai Buloh Campus, Selangor, Malaysia, 2 Hospital Orang Asli, Gombak, Kuala Lumpur, Malaysia

* mariammd@uitm.edu.my

## Abstract

### Background

The indigenous people of Peninsular Malaysia, locally known as the Orang Asli, are vulnerable to outbreaks and novel threats such as Disease X. This is due to socio-economic challenges, limited healthcare access, and exposure to zoonotic spillover. Outbreaks in this community often result in fatalities, yet their interest in prepared-ness is low and no tailored intervention exists. This study describes the development of X-SIAGA, an intervention to improve household-level preparedness for Disease X and outbreaks among the Orang Asli in Selangor using the Intervention Mapping (IM) approach, a systematic framework for developing theory- and evidence-based health interventions.

### Methods

The six-step IM framework was applied: needs assessment, setting program goals, selecting intervention methods, developing components, implementation planning, and program evaluation. Evaluation involved validation of X-SIAGA and its measure-ment tool, Household Outbreak Preparedness Evaluation (HOPE) questionnaire, with experts (n = 5) and community members (n = 14), and a pilot test (n = 18) to assess its acceptability and preliminary effectiveness.

### Results

Needs assessment confirmed the necessity of a tailored intervention for outbreak and Disease X household preparedness. Findings guided the development of logic models, outcomes, and objectives. Both X-SIAGA and HOPE showed high content and face validity (validity indexes 0.97 to 1.0). The pilot test showed high acceptability (83.3%), and HOPE measured a significant improvement in household

**Data availability statement:** The datasets generated and analysed during the current study are publicly available in the Mendeley Data repository at https://doi.org/10.17632/bg6j8h8j7y.1.

**Funding:** The author(s) received no specific funding for this work.

**Competing interests:** The authors have declared that no competing interests exist.

preparedness following X-SIAGA (mean difference = 0.22, 95% CI = 0.17–0.27, p < 0.001, Cohen's d = 1.99).

## Conclusion

X-SIAGA is a comprehensive, evidence- and theory-based intervention designed to improve household preparedness for Disease X and outbreaks among the Orang Asli in Selangor. It is valid, feasible, and acceptable in the community. X-SIAGA shows promise and is ready for full-scale trial evaluation and long-term assessment, with potential for adaptation in other communities.

---

## 1. Introduction

An outbreak refers to the occurrence of two or more cases of a similar infectious disease within a specific time and place, which may escalate into a disaster if it causes significant harm, disruption, or loss of life [1]. The World Health Organization introduced the term Disease X to describe a hypothetical infectious disease that is currently unidentified but has the potential to trigger a global pandemic [2]. Its causative agent, referred to as Pathogen X, is predicted to arise from zoonotic origins, highlighting the unpredictable and evolving nature of emerging threats that may compromise global health [3].

This vulnerability has been evident over recent decades. Major public health emergencies such as SARS, MERS, H1N1, Ebola, Zika, and COVID-19 have shown how infectious diseases can spread rapidly across borders, overwhelm health systems, and expose systemic weaknesses [4]. Although complete eradication of future outbreaks is unrealistic, adequate preparedness efforts can reduce their impact and help prevent escalation into disasters [5].

Preparedness is defined as a proactive process involving the development of plans, resources, and capacities necessary for an effective response and recovery during emergencies [6]. Preparedness ranges from grassroots efforts to broader governance structures, and as such, it can be addressed at three levels: individual or household, community, and institutional [7]. At the individual or household level, preparedness refers to the state of readiness of individuals and families in anticipation of an event that could negatively affect them, shaped by their knowledge, attitudes, and behaviors [8]. However, effective preparedness goes beyond merely raising awareness or promoting preventive actions. It must also equip individuals with practical skills such as developing planning strategies, reserving essential resources, and being capable of utilizing those resources to support themselves until professional assistance becomes available [9,10]. Yet, preparedness initiatives at household levels remain limited, and typically concentrate on increasing awareness of diseases and natural disasters, rather than on implementing operational strategies for outbreaks and emerging threats such as Disease X [11,12].

Globally, approximately 50% to 92% of households are underprepared for general emergencies [13]. There is limited data concerning household outbreak preparedness in

Malaysia. However, local studies have indicated that flood preparedness among Malaysians is lacking, with only 20% fully prepared [14], and an even smaller proportion among Indigenous groups, with merely 10% having prepared food and life-safety measures [15]. Based on these findings, preparedness for outbreaks and emerging threats are likely to be similarly low.

The Indigenous peoples of Peninsular Malaysia are known locally as Orang Asli. They are made up of three main ethnic groups: Negrito, Senoi, and Proto-Malay, each having six subgroups. Selangor has the third largest Orang Asli population, with the majority being Proto-Malays, particularly from the Temuan subgroup [15]. The Orang Asli remain a vulnerable population attributable to socioeconomic challenges, such as poverty, limited education, poor access to healthcare, and cultural beliefs that may hinder the acceptance of public health measures [16,17].

Many Orang Asli villages in Selangor are situated in deforested forest fringes and flood-prone areas [18,19]. These conditions increase the risk of zoonotic disease spillover [20], and outbreaks due to poor sanitation following flood disasters [21,22]. Selangor is one of Malaysia's most densely populated states [23], making these villages close to urban areas. Combined with their heightened vulnerability to outbreaks, this proximity increases the risk that emerging pathogens could rapidly spread beyond the community and lead to higher mortality rates [24,25].

Despite fatalities frequently documented in the Orang Asli community resulting from outbreaks [16], preparedness efforts in Malaysia have largely focused on flood-related emergencies rather than infectious diseases, and have targeted the general population rather than being tailored to the Orang Asli [26–28]. Participation in existing preparedness programs also remains low among the Orang Asli families, primarily due to the lack of culturally relevant approaches and interventions designed to engage them effectively [29].

Given these gaps, the Orang Asli population in Selangor faces a heightened risk of severe consequences in the event of outbreaks, especially from novel pathogens such as Disease X. Therefore, this study aims to systematically develop X-SIAGA (Kesiap**SIAGA**an Penyakit **X** dan Wabak), a culturally tailored intervention that is evidence- and theory-based for this vulnerable yet often overlooked community. Development of X-SIAGA employs the Intervention Mapping (IM) framework, with input from experts and the community to ensure validity and aims to improve household preparedness for Disease X and potential outbreaks among the Orang Asli in Selangor.

## 2. Materials and methods

This study adopted the IM framework, developed by Bartholomew et al., as a structured approach for designing theory- and evidence-based health promotion programs tailored to specific target groups [30]. Given its comprehensive methodology and prior effectiveness [31,32], the IM framework guided the development of the X-SIAGA intervention in six steps: needs assessment, setting of program goals, selection of intervention methods, development of program components, implementation planning, and program evaluation. Fig 1 provides an overview of the IM process for X-SIAGA.

Conceptualization of the study began on 30 September 2023, followed by the first step involving literature review and secondary data analysis, which did not involve human participants. Human involvement, including discussions, validation, and piloting was conducted from 14 October 2024–26 January 2025, with written informed consent obtained from all participants prior data collection.

Ethical and research conduct approvals were obtained from National Medical Research Register/Medical Research and Ethics Committee (Ref: NMRR-24–01859-NQ7), Universiti Teknologi MARA Research Ethics Committee (Ref: REC/10/2024 [PG/FB/33]), Gombak Orang Asli Hospital (Ref: [19] HOAG/LAT/600–6/01 Jld. 6), and the Department of Orang Asli Development (JAKOA) (Ref: JAKOA.PP.R.004 JLD 10 [04]).

### 2.1 Step 1: Needs assessment

An Intervention Development Committee (IDC) was formed, consisting of two public health academics, one public health physician, and one medical officer. One member is of Orang Asli descent to ensure cultural relevance. The IDC guided planning, development, and implementation of the intervention.

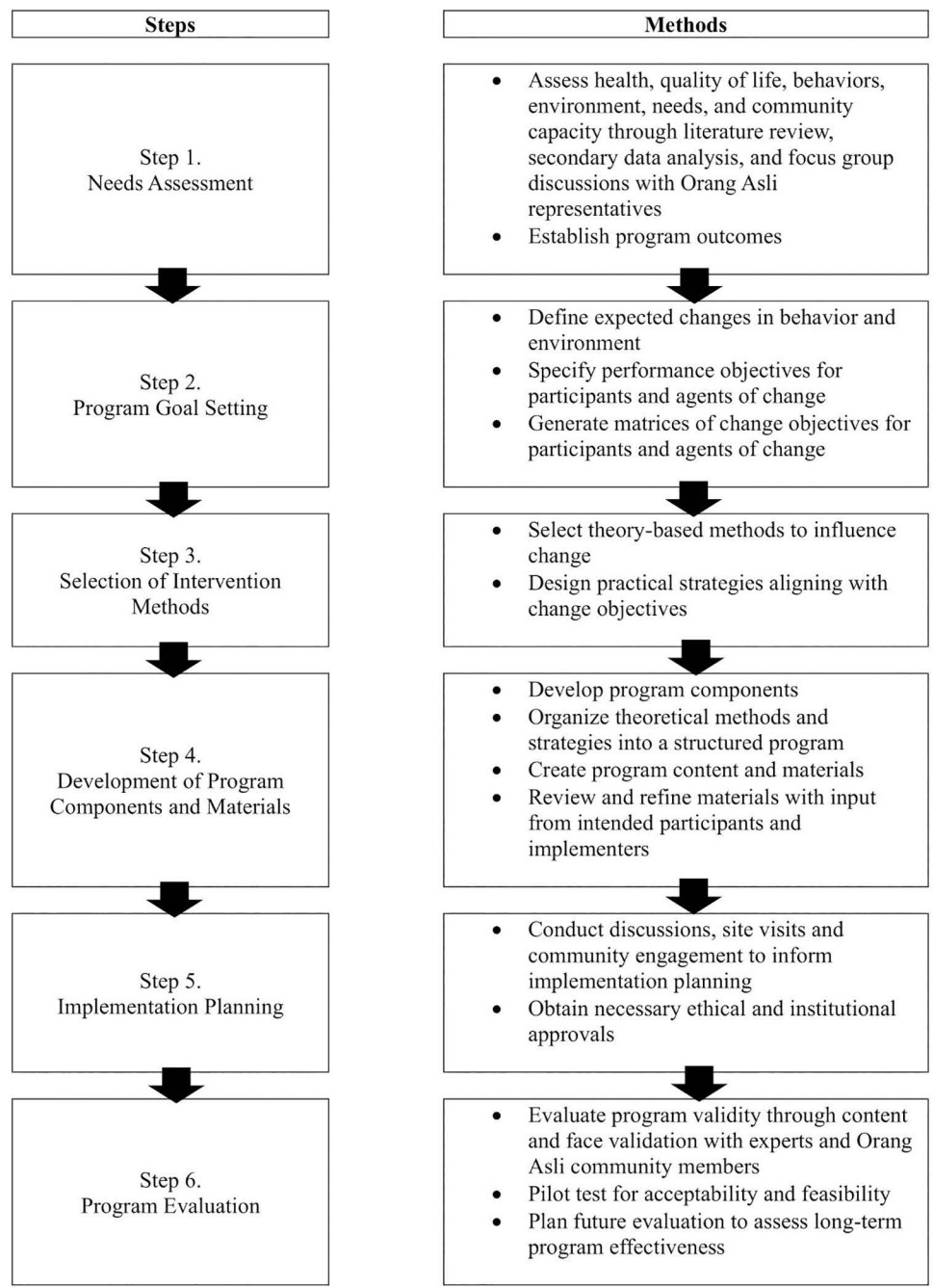

**Fig 1. Application of the intervention mapping framework in developing the X-SIAGA intervention.**

Community needs, vulnerabilities, and capacities were assessed through literature review, secondary data analysis, and focus group discussions. The literature review aimed to evaluate preparedness levels, identify unique vulnerabilities, and catalogue preparedness initiatives at individual, household, and community levels. Searches were conducted in PubMed, Google Scholar, Scopus, and Cochrane, with search terms detailed in S1 Table. No restrictions were applied

to study design or publication type given limited relevant studies. Grey literature was also included to enhance insight. Related articles in English or Malay were included, while studies on institutional preparedness were excluded. Of 425 articles identified, 45 were selected for qualitative synthesis.

Secondary analysis included publicly accessible socio-demographic data from national census records [23], Government of Selangor datasets on village locations and flood-prone areas [19], and remote sensing data on deforestation [18]. It also included deidentified infectious disease case notifications from 2000 to 2022 from the eNotifikasi database, and outbreak notifications from 2018 to 2022 from the eWabak database, which were retrieved on 22 July 2024. Authors did not have access to any individually identifiable information during or after data collection. These information were reviewed and mapped to identify Orang Asli community vulnerabilities.

Focus group discussions with five participants from the Department of Orang Asli Development (JAKOA) and Orang Asli representatives was conducted to identify specific barriers and facilitators to preparedness, including cultural practices and resource limitations. The collective insights informed a logic model of risks and outcomes, defining the aims of X-SIAGA.

## 2.2  Step 2: Program goal setting

Based on the needs assessment, matrices of change objectives were developed to align program outcomes with personal and environmental determinants. Performance objectives (POs) define the actions required of participants and agents of change to ensure effective household preparedness for Disease X and outbreaks, while change objectives specify the knowledge and behaviours necessary to achieve these actions.

POs for X-SIAGA were formulated for both participants (personal determinants) and agents of change, i.e., the program developers or implementers (environmental determinants). The participant POs were aligned with the first four levels of Bloom's Taxonomy: remember, understand, apply and analyze [33]. These POs addressed what participants must accomplish to achieve household outbreak preparedness:

i.  PO1: Participants recall and recognize Disease X and the importance of outbreak preparedness.

ii.  PO2: Participants comprehend disease transmission and associate it with the need for household preparedness for a Disease X and outbreaks.

iii.  PO3: Participants develop and apply preparedness skills and capabilities for Disease X and outbreaks.

iv.  PO4: Participants demonstrate analytical thinking, organized planning, and effective use of their household preparedness plan.

For the agents of change, complementary POs were designed to ensure the X-SIAGA intervention adequately supports participants:

i.  PO5: Ensure participants become aware of Disease X and recognize the importance of outbreak preparedness.

ii.  PO6: Guide participants in understanding disease transmission and associate it with the need for household preparedness for a Disease X and outbreaks.

iii.  PO7: Provide opportunities for participants to apply and develop preparedness skills and capabilities for Disease X and outbreaks.

iv.  PO8: Guide participants in analyzing, organizing, and practicing household preparedness for Disease X and outbreaks.

The Social Cognitive Theory (SCT) and Health Belief Model (HBM) are widely applied in preparedness programs [34]. These theories have also been combined in previous studies in developing health interventions for families and individuals

[35,36]. Formulation of X-SIAGA's change objectives for each PO was guided by both SCT and HBM, with perceived self-efficacy serving as a conceptual link between them. To achieve the POs, nine theoretical determinants were identified. These include social factors (SCT); mastery experiences, vicarious learning, verbal and social persuasion, and physiological and emotional states (from perceived self-efficacy in HBM and the cognitive factor in SCT); and perceived susceptibility, severity, benefits, and barriers (HBM).

## 2.3 Step 3: Selection of intervention methods

Intervention methods were chosen based on the theoretical framework and change objectives. The IDC brainstormed practical strategies and matched them to theoretical determinants to ensure that each strategy addresses the intended behavioural and cognitive outcomes.

## 2.4 Step 4: Development of program components and materials

The lessons in X-SIAGA were developed through input from the IDC, expert panel, and Orang Asli representatives. These lessons form the intervention, aiming to create change by addressing personal and environmental determinants through specific performance objectives. A logic model is produced to map how the intervention leads to the expected outcomes. This process guided the IDC in creating five main components of X-SIAGA:

1. Lectures on Disease X, outbreaks, and preparedness to build foundational knowledge.

2. Video presentations of past outbreaks and experience sharing to enhance understanding and shape attitudes.

3. Hands-on sessions to develop practical preparedness skills.

4. Simulation exercises to role-play real-life scenarios and strengthen application of skills.

5. Game-based learning using a card game called Wabak X, designed to reinforce key concepts.

All of the X-SIAGA intervention materials were developed by the IDC, reviewed and refined in collaboration with JAKOA officials and Orang Asli representatives from Step 1. Based on their feedback and preferences, adjustments were made to the materials.

## 2.5 Step 5: Program implementation planning

To ensure effective implementation of X-SIAGA, the IDC collaborated with JAKOA officials responsible for Orang Asli villages, reviewing existing public health initiatives and assessing feasibility, acceptability, and community resources. Site visits and engagement with village leaders and committee members helped build trust and support. Formal approval from JAKOA and clearance from the institutional ethics committee were obtained to secure access to the villages and ensure ethical conduct.

## 2.6 Step 6: program evaluation

The 32 X-SIAGA materials and the 45-item Household Outbreak Preparedness Evaluation (HOPE) questionnaire were reviewed by a panel of experts and a group of community members for content and face validity. The HOPE questionnaire, which was adapted from previous studies [37–40], was designed to evaluate the effectiveness of X-SIAGA in improving household preparedness for Disease X and other outbreaks.

The HOPE questionnaire measures two domains: Cognitive Preparedness and Preparedness Behaviour. Items are rated using multiple-choice 0–2 or 1–5 Likert scales, and missing responses were resolved at the point of administration. The HOPE instrument's psychometric properties, including content and face validation, reliability, and factor analyses, were comprehensively assessed. This paper reports only the content and face validation; assessment of other

psychometric properties was conducted in a larger validation study, and its scoring methods and are described in a separate validation paper currently under review.

For content validation, five experts reviewed the X-SIAGA materials and HOPE items for relevance to program objectives and clarity. They were selected based on relevant professional qualifications and at least two years of experience in public health, zoonotic diseases, behavioural research involving Orang Asli, or Orang Asli health and programs. The expert panel consisted of a public health physician experienced in Orang Asli healthcare, a veterinary public health specialist, a researcher with a background in behavioural research involving Orang Asli, and two JAKOA officials.

Face validation involved 14 Orang Asli community members who volunteered to participate, from villages that will not take part in the larger X-SIAGA trial. Participants included leaders, committee members and villagers with varied educational backgrounds. They assessed X-SIAGA materials for clarity, comprehensibility, and community relevance, and reviewed HOPE items for clarity and comprehensibility.

To assess the content and face validity of X-SIAGA and HOPE, a self-administered questionnaire with a 4-point Likert scale was distributed to the participating experts and Orang Asli community members (1 = disagreement, 2 = agreement with major revision, 3 = agreement with minor revision, 4 = full agreement). Written comments were also collected. For each X-SIAGA material and HOPE item, the Item-Content Validity Index (I-CVI) and Item-Face Validity Index (I-FVI) were calculated as the scale-level content validity index using the average method (S-CVI/Ave) and the scale-level face validity index using the average method (S-FVI/Ave) were calculated as the average proportion of raters who assigned a score of 3 or 4 to each content and item. A S-CVI/Ave threshold of 1.0 and an S-FVI/Ave threshold of at least 0.83 were set as the minimum for acceptability [41,42]. Scale-level content validity index based on the universal agreement method (S-CVI/UA) and scale-level face validity index based on the universal agreement method (S-FVI/UA) calculated as the proportion of contents and items on the scale that achieve a rating score of 3 or 4 by all raters. Universal agreement (UA) was defined as a score of 1, assigned when all raters were in agreement, and 0 otherwise [43,44].

Pilot testing of X-SIAGA was conducted in two Orang Asli villages using two sets of adapted questionnaires. One assessed acceptability of healthcare interventions [45], while HOPE measured preparedness levels before and immediately after X-SIAGA. Paired t-tests were used for pre–post comparisons, as they are robust to minor deviations from normality. A larger cluster randomized trial is planned in different villages to evaluate the effectiveness of X-SIAGA in longer term with follow-up HOPE assessments. For this main trial, the calculated sample size is 96 households; therefore, at least 10% of this number was targeted for the pilot [46]. Recruitment was conducted through open advertisement by village leaders. Eligible participants were Orang Asli of Malaysian citizenship, aged 18 years or older, who were heads of household or primary decision-makers for household health matters, lived in households with more than two members, and were able to converse, read, and write in Malay. Individuals who declined consent or were not permanent residents were excluded. Small incentives in the form of food baskets were provided to participants in the face validation and pilot phases; no incentives were provided to content validation experts.

## 3. Results

### 3.1 Step 1: Needs assessment

Needs assessment revealed the need of a tailored household-level intervention program on preparedness for Disease X and outbreaks. The evidence table for the literature review and the spatial mapping of Orang Asli vulnerability are provided in S2 File. In summary, Orang Asli families show low levels of disaster preparedness [29], raising concerns about their ability to respond to outbreaks and emerging threats. Key contributing factors to their vulnerability include undernutrition, low education, unstable income, poor health-seeking behaviours, reliance on traditional beliefs, distrust of formal healthcare, and reliance on forest resources or residence in deforested areas, which increase the risk of human-wildlife contact [16,47–50]. Most interventions focus on floods and seldom address Orang Asli-specific needs or emerging risks

such as Disease X [27,29]. These interconnected factors increase vulnerability to outbreaks and deepen existing inequalities [16,51].

The needs assessment produced a logic model of risks that outlines the main public health challenge, the contributing behavioural and environmental factors, the underlying determinants, and the impacts on the community, which define the intended outcomes of X-SIAGA (Fig 2).

Based on the needs assessment, the IDC determined that the program should focus on improving household preparedness for Disease X and potential outbreaks among the Orang Asli. The objective is to enhance not only knowledge and attitudes but also preparedness behaviors. Outcomes are divided into personal and environmental. The primary personal outcome is increased household outbreak preparedness. Secondary personal outcomes focus on cognitive aspects, specifically knowledge and attitudes, as well as behaviors, particularly tangible preparedness practices. For environmental outcomes, the goal is to provide evidence to justify resource allocation by stakeholders based on the X-SIAGA intervention's effectiveness.

## 3.2 Step 2: Program goal setting

The needs assessment and defined intervention outcomes in Step 1 resulted in the development of POs and matrices of change objectives. The resulting matrices for the participants (Table 1) and agents of change (Table 2) detail the necessary knowledge, skills, and behaviors to achieve the X-SIAGA outcomes.

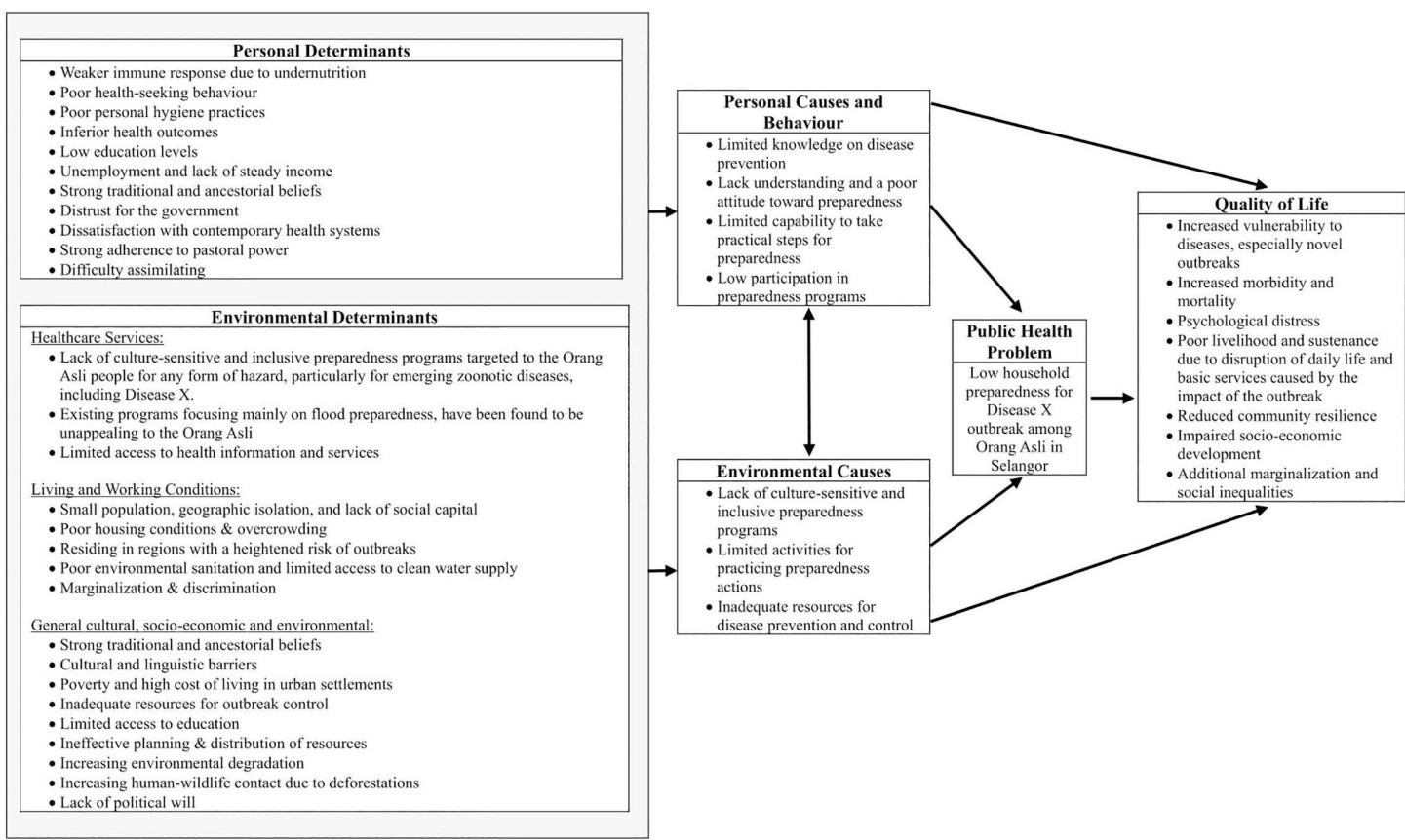

**Fig 2. Logic model of risks and problem.**

**Table 1. The matrix of change objectives for the participants (i.e., at-risk group).**

| Performance Objectives (participants) | Determinants | | | | | | |
|---|---|---|---|---|---|---|---|
| | **Mastery Experiences** | **Vicarious Learning** | **Physiological and Emotional States** | **Perceived susceptibility** | **Perceived severity** | **Perceived benefits** | **Perceived barriers** |
| PO1: Participants recall and recognize Disease X and the importance of outbreak preparedness | M1. Recall key facts about Disease X and the importance of household preparedness. | V1. Attend the program and observe role models in the community and other households participating in preparedness programs. | E1. Identify and recognize negative emotions (e.g., fear, anxiety, stress) that may arise during outbreak situations. | U1. Recognize their personal risk of infection from Disease X and outbreaks. | C1. Recognize the serious impacts of being unprepared during a potential novel outbreak. | B1. Acknowledge how preparedness actions can reduce risks and protect health. | H1. Identify potential preparedness challenges their household may face. |
| PO2: Participants comprehend disease transmission and associate it with the need for household preparedness for a Disease X and outbreaks | M2. Summarize common mistakes from previous outbreaks and personal experiences with outbreaks. | V2. Watch and reflect on stories of past outbreaks, share or listen to others' experiences, and connect these to an understanding of disease transmission and attitudes toward Disease X and outbreak preparedness. | E2. Express and predict the negative emotions they may face during an outbreak based on lessons from past outbreak stories. | U2. Describe how infectious diseases could spread in their households, summarize personal risks of Disease X infection and outbreaks, and trace which household members are most susceptible. | C2. Estimate and explain the potential negative impacts their household could face due to inadequate preparedness. | B2. Predict and infer how preparedness reduces risks and protects not only oneself but also the household and community from Disease X and outbreaks. | H2. Summarize potential obstacles to preparedness and discuss potential solutions. |
| PO3: Participants develop and apply preparedness skills and capabilities for Disease X and outbreaks | M3. Learn and practice preparedness skills (e.g., assembling kits, recognizing symptoms), use the provided templates (e.g., checklists, fillable templates), and develop their own household preparedness plan. | V3. Participate in group sessions to practice by playing simple program games with other participants and demonstrating skills | N/A | N/A | N/A | N/A | N/A |
| PO4: Participants demonstrate analytical thinking, organized planning, and effective use of their household preparedness plan | M4. Organize and demonstrate the preparedness steps learned from the program during simulation exercises, and card game (which participants are encouraged to repeat at home with their families). | V4. Participate in simulation exercises and demonstrate how to integrate household preparedness plans during a Disease X and outbreaks. | N/A | U4. Determine household outbreak risks through simulation exercises and game-based learning. | C4. Determine the negative consequences of inadequate preparedness through simulation exercises and game-based learning. | B4. Connect lessons from simulation exercises to Disease X and outbreak preparedness and outline the benefits of preparedness. | H4. Discuss and reflect on barriers and mistakes, and identify and address gaps in their preparedness plans. |

## 3.3 Step 3: Selection of intervention methods

Practical strategies for X-SIAGA intervention methods were informed by SCT and HBM and aligned with the change objectives. Table 3 outlines these strategies, which were used to design the intervention components and materials.

**Table 2. The matrix of change objectives for the agents of change (i.e., environmental agent).**

| Performance Objectives (agents of change) | Determinants | | | | | | | | |
|---|---|---|---|---|---|---|---|---|---|
| | Social Factors | Mastery Experiences | Vicarious Learning | Verbal and Social Persuasion | Physiological and Emotional States | Perceived susceptibility | Perceived severity | Perceived benefits | Perceived barriers |
| PO5: Ensure participants become aware of Disease X and recognize the importance of outbreak preparedness | S5. Involve Orang Asli representatives in program planning to ensure culturally appropriate content that helps participants recognize key messages. | M5. Repeat Disease X facts and the importance of outbreak preparedness using simple terms to help participants retain the message. | V5. Encourage village leaders and the community to participate in the program to ensure they recognize its importance. | R5. Encourage village leaders and the community to promote the program to help others recognize its importance. | E5. State the negative emotions that may arise during outbreak situations (e.g., fear, anxiety, stress). | U5. State the personal risk of being affected by outbreaks. | C5. State the serious consequences of being unprepared for novel outbreaks. | B5. State the advantages of being prepared for novel outbreaks. | H5. State the potential barriers to preparedness. |
| PO6: Guide participants in understanding disease transmission and associate it with the need for household preparedness for a Disease X and outbreaks. | S6. Adapt language and materials to culturally appropriate formats, ensuring participants relate to and comprehend the content. | M6. Explain similarities of past outbreaks repeatedly, helping participants interpret and summarize common mistakes. | V6. Illustrate lessons from past outbreaks to help participants develop understanding of disease transmission and develop better attitudes to Disease X preparedness. | R6. Facilitate discussions and encourage participants to share and reflect on their past outbreak experiences. | E6. Evoke emotions through stories, helping participants associate these feelings with the need for Disease X and outbreak preparedness. | U6. Describe individual risks of Disease X through past experiences to help participants identify and relate to their vulnerabilities. | C6. Use examples of past outbreaks to demonstrate the negative consequences of inadequate preparedness, enabling participants to understand the stakes. | B6. Highlight the benefits of preparedness using past stories to help participants compare outcomes and grasp their importance. | H6. Discuss barriers to preparedness using personal and shared experiences to help participants examine and understand obstacles they might face. |
| PO7: Provide opportunities for participants to apply and develop preparedness skills and capabilities for Disease X and outbreaks | S7. Build positive relationships with the community to support participants in applying preparedness skills. | M7. Teach participants specific skills (e.g., assembling kits, recognizing symptoms) and provide simple, easy-to-follow structures (e.g., checklists, fillable templates) that participants can follow independently. | V7. Conduct group sessions where participants demonstrate and apply their skills through simple games. | R7. Give feedback and encouragement as participants apply their skills during activities. | N/A | N/A | N/A | N/A | H7. Design simple, low-cost activities that participants can apply in real-life settings, and deliver the program to participants at their convenience. |

*(Continued)*

**Table 2.**  (Continued)

| Performance Objectives (agents of change) | Determinants | | | | | | | | |
|---|---|---|---|---|---|---|---|---|---|
| | Social Factors | Mastery Experiences | Vicarious Learning | Verbal and Social Persuasion | Physiological and Emotional States | Perceived susceptibility | Perceived severity | Perceived benefits | Perceived barriers |
| PO8: Guide participants in analyzing, organizing, and practicing household preparedness for Disease X and outbreaks | S8. Conduct simulation exercises in participants' preferred time, place, and language to model Disease X scenarios, identify symptoms, and demonstrate preparedness steps. | M8. Provide opportunities for participants to organize and demonstrate preparedness steps through simulation exercise and game-based learning. | V8. Use simulation exercises to guide participants in demonstrating the integration of their household plans during a Disease X or outbreak simulation. | R8. Recognize participants' efforts by offering rewards to acknowledge their ability to complete and apply what they've learned. | E8. Incorporate emotional elements during the simulation exercises to help participants infer potential real-life scenarios in a Disease X or outbreak. | U8. Facilitate simulation exercises and game-based learning sessions to enable participants to determine household outbreak risks, encourage for repeated play at home with their families. | C8. Facilitate simulation exercises and game-based learning sessions to enable participants to determine the consequences of inadequate preparedness, encourage for repeated play at home with their families. | B8. Encourage participants to connect lessons from simulation exercises to Disease X and outbreaks by outlining the benefits of preparedness. | H8. Discuss and reflect on barriers and mistakes during reflection sessions to help participants identify and address gaps in their preparedness. |

## 3.4  Step 4: Development of program components and materials

A logic model of intervention was created to illustrate how X-SIAGA's resources and activities lead to targeted outcomes (Fig 3). Table 4 details X-SIAGA contents and activities, the gameplay cards associated with the intervention contents, as well as the questionnaire items used for evaluation of that intervention content, and the foundational theory behind that content.

## 3.5  Step 5: Program implementation planning

Based on the feedback and insights gathered, a decentralized implementation strategy was chosen. The X-SIAGA interventionwill be rolled out within participants' villages, prioritizing accessibility and convenience by eliminating the need for participants to travel outside their communities. It will be delivered during in a one-day community workshop involving all participating households, although evaluation will be conducted at the household level rather than the community level. Scheduling will be coordinated in advance in collaboration with JAKOA officials and village leaders to encourage high levels of participation and enhance the likelihood of interventionsuccess.

## 3.6  Step 6: Program evaluation

For content validation, X-SIAGA materials and HOPE items achieved S-CVI/Ave of 1.0 (S-CVI/UA = 1.0) for relevance to intervention objectives and clarity. For face validation, Orang Asli raters gave X-SIAGA materials an S-FVI/Ave of 0.97 (S-FVI/UA = 0.81) for clarity and comprehensibility, and S-FVI/Ave of 0.99 (S-FVI/UA = 0.93) for community relevance. HOPE items scored S-FVI/Ave of 0.98 (S-FVI/UA = 0.79) for clarity and S-FVI/Ave of 0.98 (S-FVI/UA = 0.77) for comprehensibility. Rater backgrounds and detailed results are provided in S3 File.

The expert panel suggested minor wording changes for improved clarity. They noted that the hands-only Cardiopulmonary Resuscitation (CPR) session is indeed appropriate for the general public, given the importance of basic lifesaving skills in outbreak and emergency settings. However, the panel emphasized the need to present it in an accessible, reassuring manner to avoid discouraging participants. They also highlighted the need for a suitable delivery pace and interviewer-assisted HOPE questionnaire as an option to support comprehension.

**Table 3.  Theoretical determinants, methods and practical strategies.**

| Theoretical determinants | Change objectives | Theory-based methods | Practical strategies |
|---|---|---|---|
| Mastery experience | • M1 and M5: Recall Disease facts.<br>• M2 and M6: Review past outbreaks.<br>• M3: Practice preparedness skills.<br>• M3: Follow easy planning guides.<br>• M4: Apply and exhibit strategies.<br>• M7: Demonstrate readiness actions.<br>• M8: Emphasize importance repetitively. | Social cognitive theory and perceived self-efficacy | • Conduct lectures on Disease X's key facts and the significance of household outbreak preparedness to establish a solid knowledge base.<br>• Use examples from previous outbreaks and personal experiences to highlight the necessity of learning from past mistakes, deepen understanding and nurture positive attitude towards preparedness.<br>• Utilize easy-to-use tools, such as checklists and customizable plans to simplify the development of practical outbreak preparedness skills.<br>• Integrate interactive techniques, like simulation exercises to practice preparedness skills, and game-based learning that allows repeated play to enhance engagement and promote involvement of the whole family. |
| Vicarious Learning | • V1 and V5: Watch and learn from role models.<br>• V2 and V6: Analyze and discuss past outbreak narratives.<br>• V3 and V7: Practice skills in group settings.<br>• V4 and V8: Demonstrate learned actions in simulations. | Social cognitive theory and perceived self-efficacy | • Involve community leaders in the program to showcase role model behaviors and encourage participation.<br>• Use movies, documentaries, or narrated experiences of past outbreaks, then hold discussions to link what is observed to preparedness actions.<br>• Do practical sessions in groups so participants can observe and learn peers' approaches to preparedness, and be motivated to follow the good practices.<br>• Conduct simulation exercises to allow participants to witness, role-play and replicate effective outbreak preparedness and response strategies. |
| Verbal and Social Persuasion | • R5: Promote program importance.<br>• R6: Encourage discussions<br>• R7: Provide supportive feedback.<br>• R8: Acknowledge and reward participant efforts. | Social cognitive theory and perceived self-efficacy | • Engage village leaders and community advocates to actively promote the program to community members.<br>• Offer continuous feedback and encouragement throughout the program's activities to build confidence in their abilities and refine their skills.<br>• Reward the participation and efforts of participants to motivate them towards continued engagement and practice. |
| Physiological and Emotional States | • E1 and E5: Identify negative emotions during outbreaks.<br>• E2: Predict outbreak-related emotions.<br>• E6: Associate feelings with preparedness.<br>• E8: Incorporate emotions in simulations. | Social cognitive theory and perceived self-efficacy | • Deliver lectures on the range of negative emotions commonly experienced during outbreaks, such as fear, anxiety, and stress, to educate participants on the emotional landscape of outbreak situations.<br>• Use narratives and personal accounts from past outbreaks to evoke specific emotions, enabling participants to connect emotionally with the importance of being prepared for Disease X and outbreaks.<br>• Integrate emotional elements within simulation exercises to induce a more immersive understanding of how such feelings may manifest in real-life outbreak scenarios. |
| Perceived Susceptibility | • U1 and U5: Acknowledge personal Disease X risk.<br>• U2 and U6: Understand household disease spread.<br>• U4 and U8: Determine potential outbreak risks in the household. | Social cognitive theory and Health beliefs | • Conduct lectures to highlight community vulnerability and increase awareness of the personal and collective risk of Disease X infection and outbreaks.<br>• Share stories and experiences of past outbreak cases to make the risk more tangible through real-life scenarios.<br>• Use interactive techniques such as simulation exercises and game-based learning to demonstrate household vulnerabilities to Disease X and outbreaks. |
| Perceived Severity | • C1 & C5: Recognize consequences of inadequate preparedness for outbreaks.<br>• C2 and C6: Understand household impact from lack of preparedness.<br>• C4 and C8: Determine potential impacts of inadequate preparedness to the household. | Social cognitive theory and Health beliefs | • Offer lectures detailing the potential severity of a Disease X and outbreaks.<br>• Draw upon historical outbreak data to illustrate the negative outcomes of unpreparedness.<br>• Implement interactive learning methods, such as simulation exercises and game-based learning, to educate participants on the severe implications of Disease X and outbreaks when unprepared. |
| Perceived Benefits | • B1 and B5: Recognize benefits of preparedness.<br>• B2 and B6: Understand communal protection benefits.<br>• B4 and B8: see the whole picture of benefits of outbreak preparedness for the household | Social cognitive theory and Health beliefs | • Provide a lecture on how preparedness can reduce the risk and impact of Disease X and outbreaks.<br>• Use historical outbreak scenarios to illustrate how being well-prepared can lead to better health outcomes compared to being unprepared.<br>• Engage participants using simulations and game-based learning to demonstrate the tangible benefits of preparedness actions on personal, household, and community health. |

*(Continued)*

**Table 3.** (Continued)

| Theoretical determinants | Change objectives | Theory-based methods | Practical strategies |
|---|---|---|---|
| Perceived Barriers | • H1, H2, H5 and H6: Examine household preparedness challenges.<br>• H7: Design simple and low-cost tools.<br>• H8: Reflect on and address readiness inadequacies. | Social cognitive theory and Health beliefs | • Utilize past outbreak experiences to facilitate discussions on common and unique barriers to household preparedness and discuss potential solutions.<br>• Introduce low-cost, practically achievable exercises that can fit into participants' daily lives, for example like correct hand-washing technique. |
| Social Factors | • S5 and S7: Collaborate and respects cultural norms and values.<br>• S6 and S8: Tailor communication and materials | Social cognitive theory | • Collaborate with Orang Asli representatives during program development to ensure cultural sensitivity and language clarity, making the content relatable and comprehensible.<br>• Build connections with Orang Asli representatives to facilitate community engagement and participation.<br>• Embed culturally specific concepts and practices into educational materials, simulations, and games to enhance relatability and foster a deeper emotional connection with the content. |

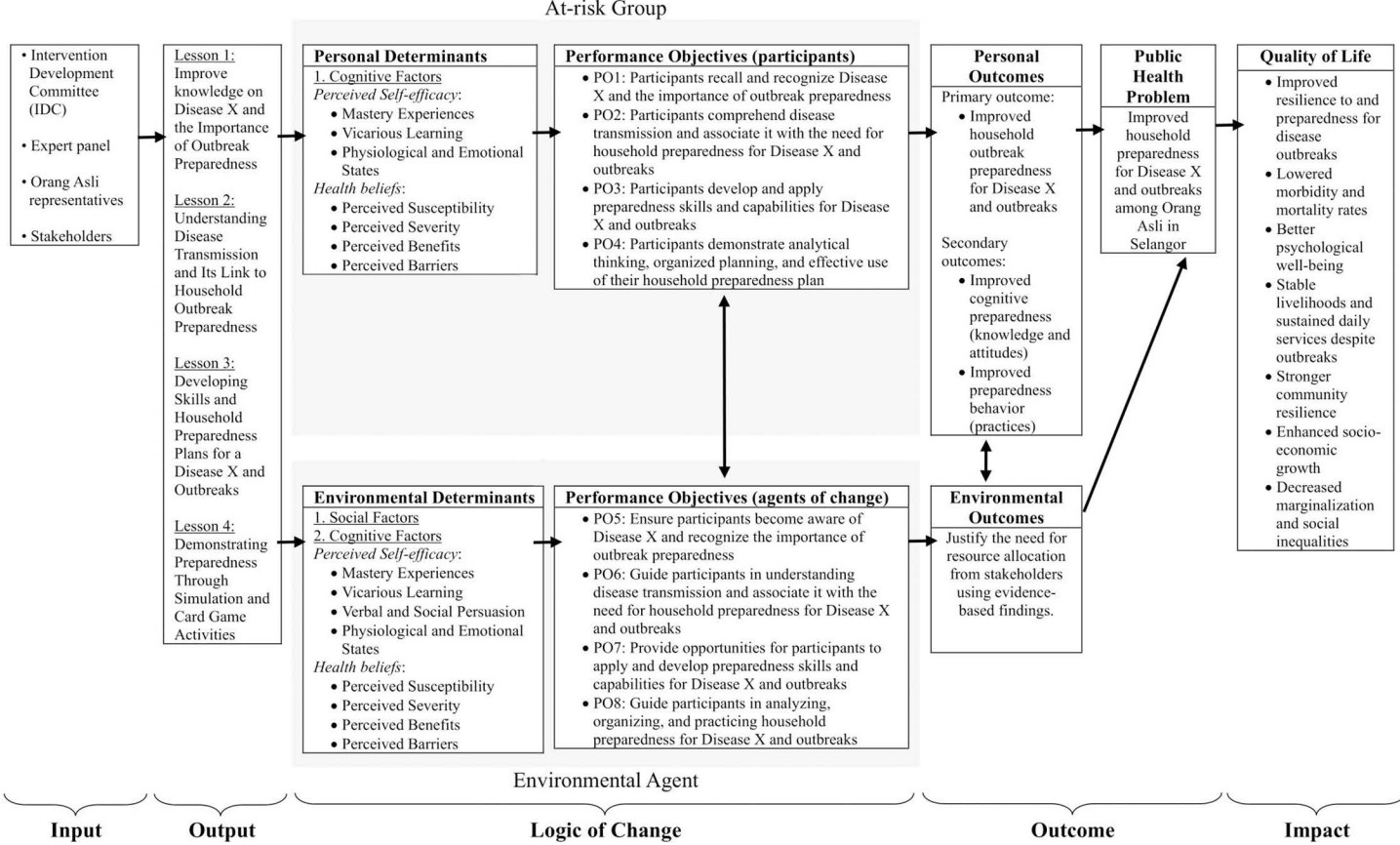

**Fig 3. Logic model of intervention.**

**Table 4. Overview of X-SIAGA intervention components and activities, linked to corresponding Wabak X game cards, HOPE questionnaire items, and the foundational theory.**

| Domain | Module (PO) | Intervention (estimated duration) | | Associated Game Card from Wabak X | Measuring Items from HOPE questionnaire | Theories |
|---|---|---|---|---|---|---|
| | | Component | Contents/Activities | | | |
| Cognitive Preparedness (Knowledge and Attitude) | Lesson 1: Improve knowledge on Disease X and the Importance of Outbreak Preparedness (PO1: Ensure participants become aware of Disease X and recognize the importance of outbreak preparedness) | Lecture (30 minutes) | 1.1 Lecture: Introduction to Disease X What is Disease X (15 minutes): • Why do we need to be concerned about Disease X • Where could Disease X start to spread • When: Situations or conditions under which Disease X is more likely to spread • How Disease X may spread and cause an outbreak • Who are the people higher risk of Disease X 1.2 Lecture: Importance of Household-Level Outbreak Preparedness (15 minutes): • What is household outbreak preparedness. • When: Timing for performing preparedness actions. • Why Preparedness Is Important for Your Family • Who is responsible for preparedness • How to Prepare for Disease X Outbreak | i. Event card: • Disease X ii. Action cards: • Isolation Order • Animal • Good hygiene iii. Item cards: • Fever medications • Wound care items • PPE • Antimicrobial agents • Important information sheet iv. Preparedness cards | K1: Knowledge of what Disease X is. K2: Knowledge of where Disease X occurs. K3: Knowledge of how Disease X spreads. K4: Knowledge of potential symptoms of Disease X. K5: Knowledge of what constitutes an outbreak. K6: Knowledge of when an outbreak can occur. K7: Knowledge of the severity of a Disease X outbreak. K8: Knowledge of outbreak responsibilities. K9: Knowledge of the definition of household preparedness for a Disease X outbreak. K10: Knowledge of items needed for Disease X and outbreak preparedness. K11: Knowledge of actions required for Disease X and outbreak preparedness. K12 – K14: Knowledge of risks associated with activities during Disease X and outbreaks. K15, K16: Knowledge of who is at risk during Disease X and outbreaks. | Social cognitive theory (cognitive factors) |
| | Lesson 2: Understanding Disease Transmission and Its Link to Household Outbreak Preparedness (PO2: Guide participants in understanding disease transmission and associate it with the need for household preparedness for a Disease X outbreak) | Video presentation and personal experience sharing (60 minutes) | 2.1 Video session and discussion (30 minutes) • Nipah Virus Outbreak • Ebola Virus Outbreak • COVID-19 Pandemic 2.2 Sharing personal experiences with past outbreaks and discussion (30 minutes) | i. Event card: • Disease X ii. Action cards: • Animal • Good hygiene • Isolation Order • Isolation Violation • Public Health • Sick Person • Village Leader • Shaman • Hunter • Neighbour | A1, A2: Perception of risks associated with Disease X and outbreaks. A3 – A5: Perception of the impact of Disease X outbreaks. A6, A7: Recognition of the benefits of outbreak preparedness. A8: Outlook on who should bear responsibility for outbreak preparedness. A9: Confidence in personal preparedness ability. A10, A11: Perception of barriers to Disease X and outbreak preparedness. Readiness to receive health interventions. A12, A13: Willingness to engage in health interventions for Disease X and outbreaks. A14 – A16: Perception of barriers to health interventions. | Social cognitive theory (cognitive factors), and health beliefs |

*(Continued)*

Table 4. (Continued)

| Domain | Module (PO) | Intervention (estimated duration) | | Associated Game Card from Wabak X | Measuring Items from HOPE questionnaire | Theories |
|---|---|---|---|---|---|---|
| | | Component | Contents/Activities | | | |
| Preparedness Behaviour (Practice) | Lesson 3: Developing Skills and Household Preparedness Plans for a Disease X Outbreak (PO3: Provide opportunities for participants to apply and develop skills and capabilities for Disease X outbreak preparedness) | Practical session (90 minutes) | 3.1 Pre-Outbreak Demonstration and Practice (30 minutes): • The 7-Step Hand Hygiene • Developing a Household Preparedness Plan for Disease X Outbreak 3.2 During and Post-Outbreak Demonstration and Practice (30 minutes): • Recognizing Symptoms Through a Puzzle Game • Learning the Flow of Steps Through a Sequencing Game 3.3 Hands-Only CPR (30 minutes) | i. Action cards: • Good hygiene • Isolation Order • Isolation Violation • Public Health • Sick Person • Village Leader ii. Item cards: • Fever medications • Wound care items • PPE • Antimicrobial agents • Important information sheet iii. Preparedness cards | P1 – P6: Prepared a household outbreak plan. P7 – P10: Assembled a household outbreak kit. P11: Performing the seven steps of hand hygiene. P12: Responding to a severely ill household member suspected of being infected with Disease X or facing an outbreak. P13: Performing hands-only CPR. | Social cognitive theory (behaviour) and perceived self-sufficiency |
| | Lesson 4: Demonstrating Preparedness Through Simulation and Card Game Activities (PO4: Guide participants in analyzing, organizing, and practicing Disease X outbreak household preparedness) | Simulation exercise (30 minutes) | 4.1 Role-playing on different outbreak scenarios (30 minutes): • Pre-Outbreak scenario • During outbreak scenario • Hands-only CPR scenario | i. Action cards: • Isolation Order • Isolation Violation • Public Health • Sick Person • Village Leader ii. Item cards: • Fever medications • Wound care items • PPE • Antimicrobial agents • Important information sheet iii. Preparedness cards | P12: Responding to a severely ill household member suspected of being infected with Disease X or facing an outbreak. P13: Performing hands-only CPR. | Social cognitive theory (behaviour) and perceived self-sufficiency |
| | | Game-based learning session (30 minutes) | 4.2 Wabak X card game play session (30 minutes) | Wabak X card game, a complete deck contains 4 types of cards: i. Item Cards ii. Preparedness Cards iii. Action Cards iv. Event Card | HOPE questionnaire, contains 45 items to measure household outbreak preparedness: i. Cognitive preparedness: 32 items (K1 – K13, A1 – A16) ii. Preparedness behaviour: 13 items (P1 – P13) | Social cognitive theory (behaviour), health beliefs, and perceived self-sufficiency |

Feedback from Orang Asli raters included concerns that Wabak X card game instruction leaflet might be overwhelming for some individuals. It was suggested that facilitators provide a verbal explanation of the game during sessions rather than relying solely on the leaflet. This suggestion was also echoed by an expert panel member.

All feedback was taken into consideration; however, no changes were made to the X-SIAGA intervention materials, as comments focused on delivery style rather than content, and all materials had validity indexes greater than 0.8, indicating that revision was unnecessary [42]. These suggestions were incorporated during the implementation phases, including pilot testing, with research assistants trained accordingly.

Pilot testing involved 18 households from two Orang Asli villages. On average, families had lived in their respective villages for 23 years, with six household members and a monthly household income of MYR 1,416. The pilot test showed good acceptability of the X-SIAGA intervention, with 83.3% of participants rating it as generally acceptable. Evaluation of pre- and immediate post-intervention revealed significant improvements in household outbreak preparedness, increasing from a mean score of 0.50 (SD 0.13) at baseline to 0.72 (SD 0.08) immediately after the intervention (mean difference = 0.22, 95% CI = 0.17–0.27, $p < 0.001$, Cohen's d = 1.99). The mean difference of 0.22 represents the average increase in normalized household preparedness scores following participation in X-SIAGA. Detailed results are available in S4 File.

## 4. Discussions

The X-SIAGA intervention was created in response to a needs assessment showing gaps in outbreak preparedness among the Orang Asli. The Orang Asli in Selangor was chosen because they are at high risk of zoonotic spillover near deforested forest edges in Malaysia's most populated state. X-SIAGA was developed using the IM framework to address both cognitive and behavioral aspects, with its contents aligned to specific objectives and the local context.

Validation of both the X-SIAGA intervention and its evaluation tool, HOPE, showed all contents and items had an average validity index above between 0.97 to 1.0, indicating strong agreement and high validity. While some minor disagreements were observed among the Orang Asli raters, this is expected since UA is sensitive to the number of raters and is often lower than the average validity index [42], and such disagreements are not expected to affect overall validity [52]. Pilot testing of the finalised X-SIAGA intervention demonstrated high acceptability among participants and significant improvements in household preparedness. These findings support X-SIAGA's readiness for full-scale trial evaluation and future broader implementation.

IM has been widely and successfully used to design health interventions for rural and underserved populations globally [53–55]. It has also been applied beyond intervention development; for example, one study used IM to examine how global economic policies and land privatization negatively affected Indigenous health in Suriname, aiming to inform more responsive policies [56].

Despite its strengths, the development of interventions using IM is not without challenges. Consistent with experiences reported in other IM-based programs [53–55], developing X-SIAGA was time- and resource-intensive, but necessary to ensure a well-designed intervention. Therefore, it is important for those planning to use IM in designing programs to allocate sufficient time and resources for the process.

Community engagement was also challenging, particularly in Orang Asli villages with limited exposure to external programs, as these communities tend to be more reserved and less receptive. Trust was built and participation encouraged through sustained collaboration with JAKOA and local leaders. However, it cannot be ensured that all responses were entirely honest, as some may have been socially desirable. Nevertheless, efforts were made to encourage honest feedback by ensuring confidentiality and fostering a respectful, non-judgmental environment.

From engagements with the community, it was also felt that there was some reluctance to participate in educational interventions, and even greater hesitation toward invasive interventions such as vaccination. This likely stemmed from the lack of immediate and visible benefits, making it hard for community members to see the value of such interventions.

These challenges were anticipated and proactively addressed in X-SIAGA by targeting the social component of SCT through the involvement of Orang Asli representatives, adaptation of culturally appropriate materials, and fostering of community trust. Vicarious learning (perceived self-efficacy) was promoted by engaging community leaders to lead participation, while perceived benefits (HBM) were highlighted using outbreak videos and the Wabak X card game to make benefits more relatable.

Integrating cultural values into the intervention without resorting to stereotypes presented a unique challenge. Considerable effort was devoted to designing the Wabak X card game, particularly in the illustrations. Given the cultural diversity even among different Orang Asli ethnic groups, the design underwent multiple revisions to ensure that character depictions and terminologies were appropriate and relatable across various subgroups in Selangor. This iterative process, involving collaboration between the IDC, village leaders, and the illustrator, was essential in producing culturally resonant materials.

Another limitation was the small pilot sample size, which included only 18 households. However, this represents over 10% of the 96 households planned for the actual trial, so it is considered acceptable [46]. Additionally, since X-SIAGA was developed and tested specifically among the Orang Asli in Selangor, its generalizability to other populations remains limited. Further limitations include the pre–post study design, which is subject to potential threats such as testing effects and the Hawthorne effect, where participants may change behavior simply due to being observed. The clustered village context may have influenced outcomes, and context effects cannot be ruled out. The absence of follow-up data in this study prevents evaluation of sustained change versus short-term effects; however, a larger trial with long-term follow-up is planned.

Nonetheless, X-SIAGA represents a pioneering initiative in supporting the Orang Asli, a vulnerable yet often overlooked community in Malaysia. Expert input and community feedback guided the IM process, resulting in an intervention that is both evidence-based and culturally relevant. Challenges encountered during X-SIAGA's development highlight the need for flexibility, consistency and sufficient resources when working with Indigenous communities. While tailored for the Orang Asli in Selangor, X-SIAGA could be adapted for other Orang Asli communities and the general population.

## 5. Conclusion

X-SIAGA is a household preparedness intervention for Disease X and outbreaks, tailored for the Orang Asli in Selangor and developed using the IM approach. Positive findings highlight the importance of systematic and participatory development of interventions for indigenous communities. Challenges encountered point to the need for sufficient resources and ongoing community engagement. X-SIAGA shows promise for full-scale trial evaluation, long-term assessment, and potential adaptation to other communities.

## Supporting information

**S1 Table. Search strategy and terms.** Search strategy and terms used for literature review for needs assessment.
(PDF)

**S2 File. Literature review and secondary data analysis results.** Summary of literature review and secondary data analysis findings for needs assessment.
(PDF)

**S3 File. Content and face validation results.** Summary of content and face validation for the X-SIAGA intervention and HOPE questionnaire.
(PDF)

**S4 File. X-SIAGA pilot testing results.** Acceptability and preliminary effectiveness of the X-SIAGA intervention.
(PDF)

## Acknowledgments

Utmost gratitude to the panel of experts and Orang Asli community members for their cooperation and willingness to contribute to the validation component of this study. Special thanks to the Orang Asli village leaders and committee members whose involvement was instrumental in encouraging community participation, despite the time-consuming nature of the research activities. Appreciation is also extended to JAKOA officials for their assistance in facilitating engagement with the Orang Asli community. Sincere thanks are also extended to everyone who was directly or indirectly involved in this work.

## Author contributions

**Conceptualization:** Ameerah Su'ad Abdul Shakor, Mariam Mohamad.

**Data curation:** Ameerah Su'ad Abdul Shakor.

**Formal analysis:** Ameerah Su'ad Abdul Shakor.

**Methodology:** Ameerah Su'ad Abdul Shakor, Mariam Mohamad, Khalid Ibrahim, Izandis Mohamad Sayed.

**Supervision:** Mariam Mohamad, Khalid Ibrahim, Izandis Mohamad Sayed.

**Visualization:** Ameerah Su'ad Abdul Shakor.

**Writing – original draft:** Ameerah Su'ad Abdul Shakor.

**Writing – review & editing:** Ameerah Su'ad Abdul Shakor, Mariam Mohamad, Khalid Ibrahim, Izandis Mohamad Sayed.

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
