## [Decision Letter · Decision Letter 0]

13 Nov 2025

Dear Dr. Mohamad,

Thank you for submitting your manuscript to PLOS ONE. After careful consideration, we feel that it has merit but does not fully meet PLOS ONE’s publication criteria as it currently stands. Therefore, we invite you to submit a revised version of the manuscript that addresses the points raised during the review process.

We look forward to receiving your revised manuscript.

Kind regards,

Mohamed Gamal Elsehrawy

Academic Editor

PLOS ONE

Journal Requirements:

3. In the online submission form, you indicated that “The data supporting the findings of this study are available from the authors upon reasonable request and with approval from the Medical Research and Ethics Committee (MREC) (contact: mreciir@moh.gov.my) and the Universiti Teknologi MARA Research Ethics Committee (UiTM REC) (contact: frcmedic@uitm.edu.my) .”.”

5. We note that Figure S2 in your submission contain map images which may be copyrighted. All PLOS content is published under the Creative Commons Attribution License (CC BY 4.0), which means that the manuscript, images, and Supporting Information files will be freely available online, and any third party is permitted to access, download, copy, distribute, and use these materials in any way, even commercially, with proper attribution. For these reasons, we cannot publish previously copyrighted maps or satellite images created using proprietary data, such as Google software (Google Maps, Street View, and Earth). For more information, see our copyright guidelines: http://journals.plos.org/plosone/s/licenses-and-copyright.

1. You may seek permission from the original copyright holder of Figure S2 to publish the content specifically under the CC BY 4.0 license.

Reviewers' comments:

Reviewer's Responses to Questions

**Comments to the Author**

1. Is the manuscript technically sound, and do the data support the conclusions?

Reviewer #1: Yes

Reviewer #2: Yes

2. Has the statistical analysis been performed appropriately and rigorously?

Reviewer #1: Yes

Reviewer #2: Yes

3. Have the authors made all data underlying the findings in their manuscript fully available?

Reviewer #1: Yes

Reviewer #2: Yes

4. Is the manuscript presented in an intelligible fashion and written in standard English?

Reviewer #1: Yes

Reviewer #2: Yes

Reviewer #1: Introduction

The authors effectively sets the stage by defining key concepts (outbreaks,Disease X,preparedness) and estsblishes the vulnerability of the Orang Asli population. However the link the between general outbreak preparedness and the specific focus on "Disease X" looks sightly underdeveloped and also the author should consider briefly mentioning the theoretical frameworks(SCT,HBM) that underpin the study so it prime the readers for the approach used.

Methods

The use of the Intervention Mapping framework is a major strenght and it is well-structured. The matrices in Tables 1& 2 should be looked at again , specify the selection criteria for the expert panel and the community members and clarify how the 18 households were selected.

Results

The validity indexes (S-CVI, S-FVI) are exceptionally high and strongly support the content and face validity of both the intervention and the measurement tool.However the term "significant improvement"

is accurate statistically but requires context. What was the baseline score? What is the range the HOPE scale? A mean difference of 0.22 is dificult to interpret without this.If you pre- and post-interventation mean score.

Reviewer #2: The manuscript titled Development of X-SIAGA to improve household preparedness for Disease X and outbreaks among Orang Asli communities in Selangor, Malaysia: Application of the Intervention Mapping framework. The authors provided some nice data about how X-SIAGA is an effective novel intervention strategy to inform household preparedness for disease X outbreaks among the Orang Asli communities in Selangor. Overall, the authors did a nice job in developing comprehensive evidenced based X-SIAGA to map out disease X in the communities. I have some comments and suggestions for the authors to consider in no particular orders.

1. The title is too long, consider shortening the title and retain focus.

2. In line 26-29, consider explaining intervention mapping (IM) approach and give the full meaning of HOPE, since this is their first mention.

3. There are many long sentences in lines 42–55, 67–80, 197–205 which could be summarized for clarity.

4. In Line 344-348, the authors acknowledge planned larger trial but sample size of 18 households is too small. If there is a limitation for the sample size to be limited to 18 households, the limitation should be explained here. A simple statement about the selected sample size, for instance “18 households were considered/selected for the pilot testing out of the 25 households from two Orang Asli Villages” will provide clarity about the sample size.

5. The authors made concluding statements about X-SIAGA implementation which at this stage of research is not evidence-backed up due to the number of trials. For instance: in line 413-419, the authors wrote “X-SIAGA is prepared for large scale implementation…..” this statement could be rephrased to “X-SIAGA shows promise for large-scale implementation”

6. I will suggest that the authors try as much as possible to reduce “the website link references”. The authors should look for good references from peer-reviewed publications.

**Do you want your identity to be public for this peer review?** For information about this choice, including consent withdrawal, please see our For information about this choice, including consent withdrawal, please see our Privacy Policy .

Reviewer #1: No

Reviewer #2: No

---

## [Author Response · Author response to Decision Letter 1]

9 Dec 2025

Respected Reviewers,

We would like to submit our revised manuscript, “Development of X-SIAGA: A Disease X and Outbreak Preparedness Intervention for Indigenous Households in Selangor, Malaysia,” for consideration as an original article in PLOS ONE.

All reviewer comments have been addressed in a point-by-point response in the attached file titled ‘Response to Reviewers’, and corresponding revisions have been incorporated into the manuscript.

Sincerely,

Mariam Mohamad,

Corresponding author

---

## [Decision Letter · Decision Letter 1]

5 Jan 2026

Dear Dr. Mohamad,

We look forward to receiving your revised manuscript.

Kind regards,

Mohamed Gamal Elsehrawy

Academic Editor

PLOS One

Journal Requirements:

Reviewers' comments:

Reviewer's Responses to Questions

**Comments to the Author**

Reviewer #2: All comments have been addressed

Reviewer #3: (No Response)

2. Is the manuscript technically sound, and do the data support the conclusions?

Reviewer #2: Yes

Reviewer #3: Yes

3. Has the statistical analysis been performed appropriately and rigorously?

Reviewer #2: Yes

Reviewer #3: N/A

4. Have the authors made all data underlying the findings in their manuscript fully available?

Reviewer #2: Yes

Reviewer #3: Yes

5. Is the manuscript presented in an intelligible fashion and written in standard English?

Reviewer #2: Yes

Reviewer #3: Yes

Reviewer #2: (No Response)

Reviewer #3: Summary of recommendations (major points)

Align claims with design: Reframe “effective/ready for implementation” to feasible/acceptable with preliminary improvement, and state it is ready for full-scale trial evaluation, not implementation.

Report HOPE instrument details: Clearly describe domains/subscales, item response options, scoring, scale range, handling of missing data, and what the reported mean difference (0.22) represents. Add reliability (e.g., Cronbach’s alpha/omega) from the pilot sample.

Strengthen statistical reporting: Specify the pre–post test used, assumptions (if relevant), report baseline and post values, 95% CIs, and effect sizes for total and subscales.

Improve validity methods transparency: Provide clearer reporting of content/face validity procedures and item/material-level summaries (or accessible supplement), and justify thresholds (especially with small rater numbers).

Fix data availability compliance: Replace/augment “available on request” with a clearer sharing pathway, ideally de-identified/aggregated data + codebook in an appropriate repository if ethics permits.

Additional improvements

Reproducibility of intervention: Provide facilitator guidance (manual/scripts), training, and fidelity monitoring tools/checklists; clarify language/reading-level adaptations.

Clarify participant selection and procedures: Recruitment, inclusion/exclusion, incentives, administration mode (self vs interviewer), and exact timing between pre and post.

Expand limitations: Note pre–post threats (testing/Hawthorne), lack of follow-up durability, clustering/context effects, and that changes may reflect short-term knowledge rather than sustained behavior change.

Minor polish: Ensure consistent terminology and briefly justify potentially non-obvious components (e.g., CPR) and address any acceptability concerns.

**Do you want your identity to be public for this peer review?** For information about this choice, including consent withdrawal, please see our For information about this choice, including consent withdrawal, please see our Privacy Policy .

Reviewer #2: No

Reviewer #3: **Yes:** Vaikunthan RajaratnamVaikunthan Rajaratnam

---

## [Author Response · Author response to Decision Letter 2]

8 Jan 2026

All reviewer comments have been addressed point by point, with corresponding revisions incorporated into the manuscript and the Respond to Reviewer file uploaded.

---

## [Decision Letter · Decision Letter 2]

8 Feb 2026

Dear Dr. Mohamad,

Thank you for submitting your manuscript to PLOS ONE. After careful consideration, we feel that it has merit but does not fully meet PLOS ONE’s publication criteria as it currently stands. Therefore, we invite you to submit a revised version of the manuscript that addresses the points raised during the review process.

We look forward to receiving your revised manuscript.

Kind regards,

Mohamed Gamal Elsehrawy

Academic Editor

PLOS One

Journal Requirements:

Reviewers' comments:

Reviewer's Responses to Questions

**Comments to the Author**

Reviewer #4: All comments have been addressed

Reviewer #5: (No Response)

2. Is the manuscript technically sound, and do the data support the conclusions?

Reviewer #4: Yes

Reviewer #5: Yes

3. Has the statistical analysis been performed appropriately and rigorously?

Reviewer #4: Yes

Reviewer #5: Yes

4. Have the authors made all data underlying the findings in their manuscript fully available?

Reviewer #4: Yes

Reviewer #5: Yes

5. Is the manuscript presented in an intelligible fashion and written in standard English?

Reviewer #4: Yes

Reviewer #5: Yes

Reviewer #4: The revised manuscript has been reassessed following the initial round of peer review. The authors have satisfactorily addressed all substantive concerns raised previously. In Revision 2, the study design and scope have been clarified, particularly through reframing claims of effectiveness to emphasize feasibility, acceptability, and readiness for full-scale trial evaluation rather than implementation.

Methodological transparency has been strengthened. The statistical analyses are now clearly specified, including the use of paired pre–post testing, reporting of baseline and post-intervention values, 95% confidence intervals, and effect sizes. The interpretation of the HOPE questionnaire outcomes has been clarified, and the distinction between pilot evaluation and full psychometric validation is appropriately stated.

The authors have also ensured compliance with PLOS ONE’s data and ethics policies. All underlying datasets are now publicly available via a recognized repository, and detailed ethics approval information and informed consent procedures are clearly reported. The manuscript is well organized, written in clear standard English, and the conclusions are appropriately supported by the data presented.

Reviewer #5: In line 1

Development of X-SIAGA: A Disease X and Outbreak Preparedness Intervention for Indigenous Households in Selangor, Malaysia

The title could be more engaging by incorporating action-oriented language or mentioning the anticipated outcomes of the intervention, such as "enhanced health resilience" or "community empowerment.

Suggested title:

X-SIAGA: Enhancing Household Preparedness for Outbreaks of Disease X Among the Orang Asli in Selangor

• We sincerely thank the author for their valuable contribution to the study, which sheds light on critical health challenges faced by vulnerable communities.This research is vital as it not only addresses the immediate need for outbreak preparedness among the Orang Asli but also serves as a model for similar interventions in other marginalized communities, promoting health equity and resilience.

• The high content and face validity of both X-SIAGA and the HOPE questionnaire indicate strong alignment with community needs and expectations. This validity enhances confidence in the intervention's design and its potential for success in improving preparedness.

• Continuous evaluation fosters a culture of learning and adaptation, ensuring that the intervention remains relevant and effective. This commitment to evaluation demonstrates responsiveness to community feedback, enhancing trust and collaboration.

In line 130

• It is notable that the team consists of two public health academics, one public health physician, and one medical officer, along with a member of Orang Asli descent for cultural relevance. However, the absence of a community health or public health nurse is concerning, as their expertise is crucial for addressing practical implementation strategies and enhancing community engagement. Including a nurse could provide valuable insights into frontline health challenges and strengthen the team's overall effectiveness in promoting health within the community.

**Do you want your identity to be public for this peer review?** For information about this choice, including consent withdrawal, please see our For information about this choice, including consent withdrawal, please see our Privacy Policy .

Reviewer #4: No

Reviewer #5: No

---

## [Author Response · Author response to Decision Letter 3]

9 Feb 2026

Thank you to the reviewers for taking the time to review our manuscript, titled “Development of X-SIAGA: A Disease X and Outbreak Preparedness Intervention for Indigenous Households in Selangor, Malaysia.” We appreciate the reviewers’ thoughtful suggestions. However, after careful consideration and discussion, we found that the proposed revisions did not align with the primary objectives of the manuscript, and therefore no changes were made. Please find our detailed justifications to the reviewer comments in the "Respond to Reviewer 5" document.

---

## [Decision Letter · Decision Letter 3]

10 Mar 2026

Development of X-SIAGA: A Disease X and Outbreak Preparedness Intervention for Indigenous Households in Selangor, Malaysia

PONE-D-25-38025R3

Dear Author,

We’re pleased to inform you that your manuscript has been judged scientifically suitable for publication and will be formally accepted for publication once it meets all outstanding technical requirements.

Kind regards,

Mohamed Gamal Elsehrawy

Academic Editor

PLOS One

Additional Editor Comments (optional):

Reviewers' comments:

Reviewer's Responses to Questions

**Comments to the Author**

Reviewer #4: All comments have been addressed

Reviewer #5: All comments have been addressed

2. Is the manuscript technically sound, and do the data support the conclusions?

Reviewer #4: Yes

Reviewer #5: Yes

3. Has the statistical analysis been performed appropriately and rigorously?

Reviewer #4: Yes

Reviewer #5: Yes

4. Have the authors made all data underlying the findings in their manuscript fully available?

Reviewer #4: Yes

Reviewer #5: Yes

5. Is the manuscript presented in an intelligible fashion and written in standard English?

Reviewer #4: Yes

Reviewer #5: Yes

Reviewer #4: The revised manuscript has been reassessed following the initial round of peer review. The authors have adequately addressed the substantive concerns raised previously. The study design and scope are now clearly framed as a pilot evaluation focusing on feasibility and acceptability rather than definitive effectiveness.

Methodological transparency has improved, including clearer reporting of the statistical analyses, baseline and post-intervention values, confidence intervals, and effect sizes. The authors have also addressed data availability and ethics requirements, with the underlying datasets now publicly accessible and ethics approval clearly reported.

The manuscript is well organized, written in clear English, and the conclusions are appropriately supported by the data presented. I have no further major concerns.

Reviewer #5: We sincerely thank the author for the well-conducted research and for making the required revisions thoroughly and professionally

**Do you want your identity to be public for this peer review?** For information about this choice, including consent withdrawal, please see our For information about this choice, including consent withdrawal, please see our Privacy Policy .

Reviewer #4: **Yes:** Idowu Michael AriyibiIdowu Michael Ariyibi

Reviewer #5: No

---

## [Editor Report · Acceptance letter]

PONE-D-25-38025R3

PLOS One

Dear Dr. Mohamad,

I'm pleased to inform you that your manuscript has been deemed suitable for publication in PLOS One. Congratulations! Your manuscript is now being handed over to our production team.

Kind regards,

on behalf of

Dr. Mohamed Gamal Elsehrawy

Academic Editor

PLOS One